# Public Innovation Policy Evaluation: Brazilian Electricity Tariff Paid by Consumers

**Gilmar dos Santos Marques** [1,*] , **Maria Amélia de Paula Dias** [2] **and José Luiz de Andrade Franco** [1]

1   Center for Sustainable Development (CDS), University of Brasilia (UnB), Brasilia 70910-900, Brazil
2   Faculty of Economics, Administration, Accounting and Public Policy (FACE), University of Brasilia (UnB), Brasilia 70910-900, Brazil
*   Correspondence: gilmar.marx@gmail.com

**Abstract:** The objective of this study was to evaluate the impact of the SEB public policy of innovation on the average supply tariff (AST) from its implementation in the year 2000 until 2020, considering the tariff moderation foreseen in the RD&I of the SEB, which is aligned with SDG 7 of Agenda 2030. The methodological procedures included (a) a sample consisting of 40% of the market (seven of the one hundred and eleven electricity distributors) of the market (in consumers and billing); (b) a before and after real tariff model (1994–2020) applied through ordinary least squares (OLS) estimation; (c) calculation of the real AST of the sample using the present value (PV) method, updated based on the IPCA as an indexer; (d) calculation of the result of Model 4 before and after and the AST in PV to verify the impact of the SEB's RD&I public policy on the AST. The results showed that the creation of the innovation contribution (ContribInova) impacted the AST value by 60.95% in terms of real increase. Three electricity purchasing power (EPP) indicators and the electricity purchasing power index (EPP*i*) were developed and calculated to track the evolution of the AST to help guide the tariff moderation of the SEB and the Brazilian commitment to Agenda 2030 to ensure access to energy services at affordable prices by 2030.

**Keywords:** innovation public policies; SEB RD&I; AST; Agenda 2030; SDG 7





## 1. Introduction

Policymakers, when thinking of investing in a research, development, and innovation (RD&I), start from the assumption that such an intervention will positively impact the sector. An example of this is the Research & Development Program (R&DP) and the Energy Efficiency Program (EEP) of the Brazilian electricity sector (SEB), which has tariff moderation as one of its aims [1,2]. The public policy for innovation in the SEB began with Law No. 9.991 in 24 July 2000, establishing the Energy Sector Fund (CTEnerg) and later implementing the R&DP and EEP regulated by ANEEL [2,3].

Every public policy must be monitored and evaluated in the course of its existence, and this must be foreseen in its design [1]. However, in the case of the RD&I of the SEB, which involves R&DP and EEP—regardless of being regulated by ANEEL, which is still a form of monitoring and evaluation—impacts have not been evaluated, despite being a public policy that is more than two decades old.

Precedents in the literature indicate the need to evaluate these programs. For example, Jamasb and Pollitt [4–6] indicated that ultraliberal markets, such as the U.K., have not been able to solve the problems of innovation in the electricity sector. Additionally, in 31 countries of the Organization for Economic Cooperation and Development (OECD), the international standard of regulation has not found a point of balance that would allow advances in the development of technology and innovation in the electricity sector [7,8]. Thus, the RD&I of the SEB lacks an evaluation method to measure the impact of the SEB's public innovation policy.

According the Paris Agreement, to which Brazil became a party and committed to the 2030 Agenda and its 17 Sustainable Development Goals (SDGs), the need has increased to evaluate the program and check the performance of the SEB's public policy on innovation and whether it is aligned with the commitments made by Brazil under the United Nations Framework Convention on Climate Change (UNFCCC) at COP21 [9].

This study aimed to cover a gap in the literature identified in a bibliometric study [8]. This was accomplished by assessing the impact of the Brazilian public policy on the RD&I of the SEB concerning the tariff moderation foreseen in the commitments made by Brazil through the United Nations (UN) Agenda 2030 and, in particular, with SDG 7, target 7.1: "by 2030, ensure universal, reliable, modern and affordable access to energy services" [10].

Thus, our study aimed to answer the following question: What is the impact of the implementation of the public RD&I policy of the SEB, through the R&DP and the EEP, on the tariff moderation policy foreseen in the program and goals of Agenda 2030?

To answer the study question, our objective was to evaluate the impact of the Brazilian public policy on SEB innovation from its implementation in the year 2000 until 2020, on the average supply tariff (AST), because one of the premises of RD&I is tariff moderation, which aligns with SDG 7 of Agenda 2030.

The justification for this study is that the SEB public policy of innovation, the RD&I regulated by ANEEL, has been in place for 21 years, since 2020, and one of its proposals is to promote tariff moderation through innovation. As such, an impact evaluation was needed of this public policy to verify whether this objective has been achieved, especially given the need to align with the goals of Agenda 2030.

The remainder of this paper is structured in to four sections: Theoretical Framework, Methodological Procedures, Results, and Final Considerations.

## 2. Theoretical Framework

In this section, we address the basic concepts of public policy, public policy evaluation, regression analysis, and regression modeling with binary variables (dummies), innovation in the SEB and its relationship with the principle of tariff moderation, Brazil's commitments to Agenda 2030, and indicators and purchasing power index of electric power in Brazil.

### 2.1. Public Policies

A public policy is a guideline designed to address a public problem comprising two fundamental elements: (a) intentionality and (b) response to a public problem. Hence, the motivation for creating and implementing a public policy is to treat or solve a collective and relevant problem [11]. Public policy is "everything that a government decides to do or not to do" [12].

In 2000, Law 9.991/2000, which instituted the CTEnerge, paved the way for the creation and implementation of the SEB's public policy for innovation (RD&I of the SEB), represented by the R&DP and the EEP regulated by ANEEL [2,3]. The initiative was financed through the contribution for innovation (ContribInova), created to finance this public policy and collected by the SEB companies per segment, as follows: (a) distribution contributes with 1.0% of the net operating income (NOI), being 0.5% for R&DP and 0.5% for the EEP; and (b) generation and transmission contributes with 1.0% of the NO, for R&DP [2,3].

The resources for the operationalization of the RD&I of the SEB are managed following the current legislation [2,3] and according to the following parameters: (a) National Fund for Scientific and Technological Development (FNDCT) receives 40.0% of the R&DP ContribInova amount; (b) ANEEL receives 40.0% of the R&DP ContribInova amount; and (c) Ministry of Mines and Energy (MME) receives 20.0% of the R&DP ContribInova amount [2,3,13].

Thus, the aim of the public policy for SEB innovation is to address the problem of how to innovate in a complex sector that involves generation, transmission, and distribution (GTD) networked. Therefore, the action is characterized as public policy because its meets

the two basic assumptions: intentionality and being a relevant public problem, requiring the translation of purposes into programs and actions [12,14,15].

## *2.2. Public Policy Evaluation*

Public policy evaluation is a method of checking the impact of a government's actions in seeking to solve problems that affect society. The evaluation may involve what the government does and what it has fails to do [14,16–18]. The evaluation commonly occurs before (ex ante) implementation to confirm the correct diagnosis of the problem that led to the creation of public policy. In this late detection of formulation and design, error are avoided, which enables the correction of eventual deviations and prevents the waste of public resources [19]. Another method of evaluating public policy is to conduct a posteriori (ex post) evaluation, usually at the end of the policy period [20,21]. This is a valuable instrument for the decision-making process because it allows the manager to evaluate the results: whether the problem identified was addressed according to the previously established objectives and goals; whether the resources were correctly allocated; and ultimately, to determine the necessary corrections to improve the quality of public spending [21,22].

In this study, we considered an ongoing public policy after more than 20 years of implementation. At this point, an impact evaluation was required to check the results of the policy over time [1]. Authors suggested estimating the effect of a policy using the before-and-after evaluation model because it comprises three important moments of a public policy, program, or project: before, during, and after [1,22].

## *2.3. Innovation in SEB and Its Relationship to Tariff Moderation Principle and Brazil's Commitments to Agenda 2030*

We compared the results of SEB innovation, considering the principle of tariff moderation and Brazil's commitments to the 2030 Agenda, derived from the Paris Agreement. The concept of tariff moderation to be adopted is enshrined in Article 6, §1 of Law Nos. 8987 of 1995 [23,24] and is interpreted by jurists as follows: "[ ... ] having the public service to reach and satisfy the various social groups in the pursuit of the common good, [ ... ] must be consistent with the economic possibilities of the Brazilian people, that is, the lowest possible" [25]. Bandeira de Mello [26] stated that "such modicity, it should be noted, is one of the most important user rights, because, if disrespected, the service itself will end up being unconstitutionally withheld [ ... ]". Therefore, the principle of tariff modicity for this study and the Brazilian society is relevant [27].

As Brazil's commitments to Agenda 2030 should be incorporated into public policies, we set goal 7.1 within SDG 7 (Affordable and Clean Energy) as the parameter for comparison. Its objective is to "ensure access to affordable, reliable, sustainable and modern energy for all" and its ultimate goal (according to Agenda 2030, finalist goals are those whose object relates directly (immediately) to the achievement of the specific SDG [28]) is 7.1: "by 2030 ensure universal access to affordable, reliable, and modern energy services" [10,28–31].

## *2.4. Indicators and Purchasing Power Index*

We cannot manage what we do not measure [32]; therefore, building indicators to measure results is important for any management model. "Indicators are measures, that is, they are an assignment of numbers and objects, events or situations according to certain rules" [33].

We developed three indicators and an index to more effectively manage the principle of tariff moderation and target 7.1 of SDG 7 of Agenda 2030. They are aligned with the SMART concept, which is an acronym for specific (S), measurable (M), attainable (A), relevant (R), and time-bound (T) [34]. The developed indicators are based on secondary data one of which, the average income of Brazilians, is highly aggregated. Indicators are simplifications of complex realities: even if they do not exactly describe reality, they help to interpret it and contribute to evaluating the evolution of a problem [35,36].

## 3. Methodological Procedures

The methodological procedures in this study consisted of a combination of bibliographic, documentary, and experimental research methods. Through bibliographical analysis, we sought to understand the results of published studies on SEB public policy innovation. In the literature, we identified the data referring to the electricity tariffs from 1994 to 2020, the value of the contribution to the SEB innovation program in ANEEL's database in the period from 2000 to 2020, as well as the objective of SEB tariff moderation [2,37]. With the experimental study, we aimed to test the hypothesis that creating public policy and adding a mandatory contribution to finance the innovation program would produce efficiency gains for the companies and system in a way that would benefit consumers via tariff reductions. Figure 1 shows the structure of the study.

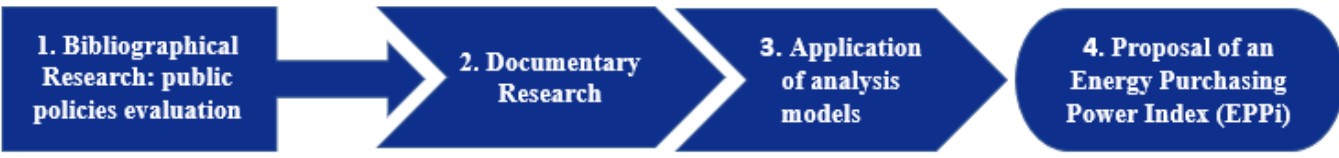

**Figure 1.** Study structure.

### 3.1. Sample Selection and Data Collection

The analysis and data survey were conducted based on the real AST/MWh value analyzed over the period from 1994 to 2022 (to project the nominal value of the AST, we opted to adopt the data projected for 2021 and 2022, because they were released by ANEEL), employing a sample of 7 out of 111 large electricity distributors. This sample accounted 6.30% of all distributors in operation in 2021 but which together commercialized 38.55% of the electricity consumed, accounting for 39.26% of the sector's revenue (without taxes) and reaching 38.73% of consumers in the national territory in all consumption classes. The justification for the selection of this sample is that these distributors operate in 4 of the 5 regions in Brazil: (a) CEMIG, ELETROPAULO, and LIGHT, from the southeast; (b) CEEE and COPEL, which operate in the south region; (c) CELG from the Midwest, and (d) CELPE from the northeast. Additionally, these companies have a history that allowed the application and evaluation before-and-after model, linear regression analysis using ordinary least squares (OLS) estimation, and they represent approximately 40.00% of the national consumption [38]. Figure 2 describes the application of the analysis models.

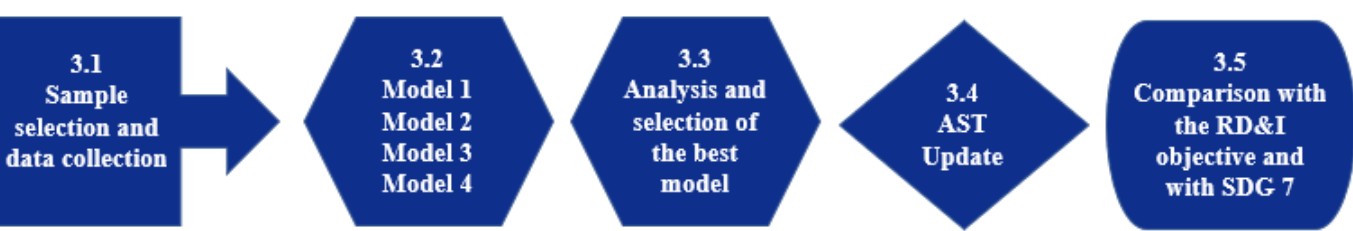

**Figure 2.** Application of the analysis models.

### 3.2. Data Analysis Methods

Linear regression analysis is "a set of tools whose focus is to make inferences, most often causal" from evidence found for a sample, which allows "making generalizations of results to the population" [39].

To conduct this experimental study, we used regression analysis. In general, regression analysis seeks to estimate the mean value of the dependent variable of a population-based on known values from the sample. In this study, we used it to estimate the effects of the exploratory variables, especially the contribution of the dummy (independent) variable to the dependent variable (energy tariff). Whereas regression analysis deals with the dependence of one variable on others, this does not necessarily imply causation. In the

words of Kendall and Stuart, "a statistical relationship, however strong and suggestive, can never establish a causal connection: our ideas of causation must come from outside statistics, ultimately from some theory [40].

The types of data available include panel data and observations of prices from various companies over time, both of which are quantifiable variables. However, other qualitative variables can affect the dependent as well as the quantitative variables, so it a binary variable, also known as a dummy variable, is commonly constructed. These variables take values of 0 and 1, which indicate the absence and the presence of the attribute, respectively [40].

Next, the before and after the real tariff model was applied through OLS estimation to identify any changes in the average prices of the AST/MWh after the establishment of the ContribInova obligation using four regression models [17,18].

a.     Model 1: Simple before-and-after regression:

$$\log(RealTariff)_{it} = \alpha + \beta_t \text{dummy}(Contribution)_t + \varepsilon_{it} \tag{1}$$

where $\log(RealTariff)_{it}$ is the value of the logarithm of the real tariff of company $i$ at time $t$ in constant 2020 prices; $\text{dummy}(Contribution)_t$ takes a value of 0 from 1994 to 1999 and 1 in the period from 2000 to 2020; and $\varepsilon_{it}$ represents the standard error of the model.

b.     Model 2: Before-and-after regression with time fixed effects

In Model 2, fixed effects were added for the time variable ($data_t$), with a dummy for each of the years analyzed. The aim was to capture movements that may have directly affected the value of the AST/MWh in a specific year. The following equation was used:

$$\log(RealTariff)_{it} = \alpha + \beta_t \text{dummy}(Contribution)_t + \theta_t data_t + \varepsilon_{it} \tag{2}$$

where $\log(RealTariff)_{it}$ is the value of the logarithm of the actual tariff for company $i$ at time $t$ in constant 2020 prices; $\text{dummy}(Contribution)_t$ takes value of 0 for 1994 to 1999 and 1 for 2000 to 2020; and $\varepsilon_{it}$ represents the standard error of the model.

c.     Model 3: Before-and-after regression with company fixed effects

The company may also have specific characteristics that influence its pricing. Thus, in Model 3, fixed effects were added for each of the companies to verify if the price levels differed from company to company, according to the following equation:

$$\log(RealTariff)_{it} = \alpha + \beta_t \text{dummy}(Contribution)_t + \mu_i company_i + \varepsilon_{it} \tag{3}$$

where $\log(RealTariff)_{it}$ is the value of the logarithm of the real fare for company $i$ at time $t$ in constant 2020 prices; $\text{dummy}(Contribution)_t$ takes value of 0 for 1994 to 1999 and 1 for 2000 to 2020; and $\varepsilon_{it}$ represents the standard error of the model.

d.     Model 4: Before-after regression with time and company fixed effects.

Finally, in Model 4, the log of the real AST/MWh was regressed as a function of the temporal dummy that indicated the implementation of the public policy that established the real contribution to innovation for SEB companies considering the presence of time and company fixed effects. Thus, with Model 4, we estimated the specific effects of each of these variables, aiming to identify differences reflecting changes in tariffs based on company $\mu_i$ at time $\theta_t$, according to the following equation:

$$\log(RealTariff)_{it} = \alpha + \beta_t \text{dummy}(Contribution)_t + \theta_t + \mu_i + \varepsilon_{it} \tag{4}$$

where $\log(RealTariff)_{it}$ is the value of the logarithm of the real tariff of company $i$ at time $t$ in constant 2020 prices; $\text{dummy}(Contribution)_t$ takes value of 0 from 1994 to 1999 and 1 from 2000 to 2020 and $\varepsilon_{it}$ represents the standard error of the model.

In Model 4, when a positive effect on tariffs was detected after the implementation of the mandatory contributions in innovation, the magnitude of the effect was verified, which we obtained through the exponential scale model. To evaluate its magnitude compared with the increase in the AST/MWh per company and percentage, we transformed the obtained value of the effect from an exponential scale (due to the model characteristics) to a percentage scale according to Equation (5):

$$effect = (\exp(\beta) - 1) \times 100 \tag{5}$$

where *effect* is the expected result of transforming the exponential scale effect to percentage; $\exp(\beta)$ is the exponential scale effect value.

### 3.3. Monetary Updating of the Average Electricity Supply Tariff (AST)

Because we considered a long period, we monetarily updated the AST/MWh and analyzed its evolution in the study period (1994 to 2022) using the present value (*PV*) method from the historical value (*HP*) of the AST/MWh and effective date until the current date, which was updated by the IPCA variation in the period being analyzed, according to IBGE (2021), with the following equation:

$$FV = PV(1+i)^n \text{ or } FV = PV(1+i)^n \tag{6}$$

where *FV* is future value; *PV* is present value or $PV_{AST}$ is the present value of AST/MWh; *i* is the inflation index (IPCA); *n* is the number of periods.

### 3.4. Comparison of Average Supply Tariff (AST) with ODS 7

Next, the real increase in the AST/MWh was compared after the implementation of the SEB innovation public policy from 2000 to 2020 (Model 4 before and after was applied between 1994 and 2020, even though it contained data for 2021 and 2022, because a regression was applied to past periods) with target 7.1 and SDG 7. We performed the comparison using the AST/MWh value (at present value) of a group of electric power distribution companies using the evolution of the Extended National Consumer Price Index (IPCA), which is calculated by the Brazilian Institute of Geography and Statistics (IBGE), in the analyzed period. This comparison was combined with an indicator: the positive effect on the AST/MWh generated by Model 4 (before and after) that occurred after the implementation of the ContribInova obligation and created by the SEB public innovation policy.

### 3.5. Creation of the Electricity Purchasing Power Index (EPPi)

Finally, we developed a set of 3 indicators and an energy purchasing power index. The electricity purchasing power (EPP) was determined as follows: first, based on the national minimum wage (NMW); second, based on the Minimum Wage Basic Food Basket of DIEESE (MWBFBDIEESE); and third, based on the average income of Brazilians, as calculated by IBGE (AIbIBGE). Below, we describe the proposed indicators, which were calculated from 1994 to 2020, except for the EPPAIbIBGE, which was calculated from 2012 to 2020.

The first electricity purchasing power indicator was calculated based on the NMW (EPPNMW) with the following formula:

$$\text{EPPNMW in MWh} = \frac{\text{Value of National MW}}{\text{Value of AST in MWh}} \tag{7}$$

The second electricity purchasing power indicator, based on the MWBFBDIEESE (EPPMWBFBDIEESE), was obtained with the following formula:

$$\text{EPPMWBFBDieese in MWh} = \frac{\text{Value of MWBFBDieese}}{\text{Value of AST in MWh}} \tag{8}$$

The third electricity purchasing power indicator, based on the AIbIBGE (EPPAIbIBGE), was determined from the following formula:

$$EPPAIbIBGE \text{ in MWh} = \frac{\text{Value of AIbIBGE}}{\text{Value of AST in MWh}} \tag{9}$$

where NMW is the national minimum wage published by the Federal Government in effect in December of each year; MWBFB is the value of the basic food basket minimum wage, published by DIEESE, in December each year; AIb is the average Brazilian income, released by the IBGE, together with the salary mass study. When not released in the current year, the previous year's value was repeated; MWh is the base value of AST/MWh without taxes for each company in December each year. The value used for these indicators was the weighted average $(\overline{X}_p)$ of the AST/MWh of the 7 concessionaries that comprised the study group for the study period.

After obtaining the partial results, the $\overline{X}_p$ of the 3 EPP indicators was calculated to generate a higher-quality input for the calculation of the electricity purchasing power index (EPP*i*). We selected the weighted average due to the larger variation in AST/MWh among the studied concessionaires as well as the variation among the EPP indicators.

$$\overline{X}_p \text{ of } EPP = \frac{\sum_{i=1}^{n} p_i \times x_i}{\sum_{i=1}^{n} p_i} \tag{10}$$

where $\overline{X}_p$ of EPP is the weighted average of the electricity purchasing power indicator (EPP); $\sum_{i=1}^{n} p_i \times x_i$ is the sum of the weighted average of the EPPNMW + EPPMWBFB/DIEESE + EPPAIb/IBGE; $\sum_{i=1}^{n} p_i$ is sum of the weights of EPPNMW + EPPMWBFB/DIEESE + EPPAIb/IBGE.

The weighting was performed from 1994 to 2011 with 2 indicators and from 2012 to 2020 with 3 indicators, after the inclusion of the EPPAIbIBGE indicator, which was first released in the year 2012.

Then, we calculated the EPP*i* from the $\overline{X}_p$ of the 3 indicators EPPNMW (MWh), EPPMWBFBDieese (MWh), and EPPAIbIBGE (MWh). Once the $\overline{X}_p$ of the 3 indicators was obtained, the evolution of the $\overline{X}_p$ of the indicators was calculated as a percentage with base 100.

$$EPPi = \left( \frac{\overline{X}_p \text{ Year } x}{\overline{X}_p \text{ Base year}} \right) \times 100 \tag{11}$$

where EPP*i* is the electricity purchasing power index; $\overline{X}_p$ *Year x* is the weighted average of the EPP indicators for the year under study; $\overline{X}_p$ *Base Year* is the weighted average of the EPP indicators for the year 1999, considering the year on a 100 basis.

In the analysis of the EPP*i* base 100, from 2000 to 2020, we considered the year 1999 (the year before ContribInova started to be charged = base year) starting in the year 2000 (year *x*) to calculate the percentage variation from one year to the other. Thus, a result equal to 100 indicated a zero effect on the AST/MWh; a variation greater than 100 indicated it was favorable to the consumer; but a value below 100 indicated that the consumer was paying more for their electricity bill, therefore losing purchasing power, as shown in Figure 3.

**Figure 3.** Creation of the electricity purchasing power index (EPP*i*).

## 4. Results and Discussion

The results and discussion of this study are divided into five sections for better understanding:

(a)   OLS estimation;
(b)   Analysis of the estimation of OLS Model 4;
(c)   Behavior of the AST/MWh (1994 to 2020) before and after ContribInova;
(d)   Evaluation of the impact of the SEB's RD&I public policy on the AST/MWh after the creation of ContribInova, including projections for the years 2021 and 2022;
(e)   Comparison of the effects of the SEB's RD&I public policy versus the goals of the 2030 Agenda SDG 7 Goal 7.1;
(f)   Development of three indicators and one index of electric power purchasing capacity.

### 4.1. OLS Estimation in 4 Models: Models 1, 2, 3 and 4

OLS estimation was performed for each of the four models (Model 1, Model 2, Model 3, and Model 4), as shown in Table 1. It is believed that Model 4 is the best since it presents the treatment for company and date or time fixed effects and has higher explanatory power, where $R^2 = 0.77246$ [41].

**Table 1.** Results of the estimated models.

|  | **Model 1** | **Model 2** | **Model 3** | **Model 4** |
|---|---|---|---|---|
| Interceptor | 5.80686 *** | 5.79582 *** | 5.84531 *** | 5.83427 *** |
|  | (−0.02296) | (−0.04403) | (−0.03357) | (−0.04311) |
| Dummy Contrib | 0.28620 *** | 0.43296 *** | 0.28620 *** | 0.43296 *** |
|  | (−0.02618) | (−0.06227) | (−0.02479) | (−0.05495) |
| 31 December 1995 |  | −0.01985 |  | −0.01985 |
|  |  | (−0.06177) |  | (−0.05433) |
| 31 December 1996 |  | −0.00501 |  | −0.00501 |
|  |  | (−0.06177) |  | (−0.05433) |
| 31 December 1997 |  | 0.00024 |  | 0.00024 |
|  |  | (−0.06177) |  | (−0.05433) |
| 31 December 1998 |  | 0.0163 |  | 0.0163 |
|  |  | (−0.06177) |  | (−0.05433) |
| 31 December 1999 |  | 0.07451 |  | 0.07451 |
|  |  | (−0.06177) |  | (−0.05433) |
| 31 December 2000 |  | −0.37595 *** |  | −0.37595 *** |
|  |  | (−0.06177) |  | (−0.05433) |
| 31 December 2001 |  | −0.30397 *** |  | −0.30397 *** |
|  |  | (−0.06177) |  | (−0.05433) |
| 31 December 2002 |  | −0.22382 *** |  | −0.22382 *** |
|  |  | (−0.06177) |  | (−0.05433) |
| 31 December 2003 |  | −0.15582 * |  | −0.15582 ** |
|  |  | (−0.06177) |  | (−0.05433) |
| 31 December 2004 |  | −0.10882 · |  | −0.10882 * |
|  |  | (−0.06177) |  | (−0.05433) |
| 31 December 2005 |  | −0.05453 |  | −0.05453 |
|  |  | (−0.06177) |  | (−0.05433) |

**Table 1.** *Cont.*

|  | Model 1 | Model 2 | Model 3 | Model 4 |
|---|---|---|---|---|
| 31 December 2006 |  | −0.03403 |  | −0.03403 |
|  |  | (−0.06177) |  | (−0.05433) |
| 31 December 2007 |  | −0.08678 |  | −0.08678 |
|  |  | (−0.06177) |  | (−0.05433) |
| 31 December 2008 |  | −0.15682 * |  | −0.15682 ** |
|  |  | (−0.06177) |  | (−0.05433) |
| 31 December 2009 |  | −0.17602 ** |  | −0.17602 ** |
|  |  | (−0.06177) |  | (−0.05433) |
| 31 December 2010 |  | −0.20035 ** |  | −0.20035 *** |
|  |  | (−0.06177) |  | (−0.05433) |
| 31 December 2011 |  | −0.22853 *** |  | −0.22853 *** |
|  |  | (−0.06177) |  | (−0.05433) |
| 31 December 2012 |  | −0.23873 *** |  | −0.23873 *** |
|  |  | (−0.06177) |  | (−0.05433) |
| 31 December 2013 |  | −0.41018 *** |  | −0.41018 *** |
|  |  | (−0.06177) |  | (−0.05433) |
| 31 December 2014 |  | −0.38041 *** |  | −0.38041 *** |
|  |  | (−0.06177) |  | (−0.05433) |
| 31 December 2015 |  | −0.05154 |  | −0.05154 |
|  |  | (−0.06177) |  | (−0.05433) |
| 31 December 2016 |  | −0.09814 |  | −0.09814 · |
|  |  | (−0.06177) |  | (−0.05433) |
| 31 December 2017 |  | −0.17820 ** |  | −0.17820 ** |
|  |  | (−0.06177) |  | (−0.05433) |
| 31 December 2018 |  | −0.06815 |  | −0.06815 |
|  |  | (−0.06177) |  | (−0.05433) |
| 31 December 2019 |  | −0.04296 |  | −0.04296 |
|  |  | (−0.06177) |  | (−0.05433) |
| CELG |  |  | −0.07555 · | −0.07555 ** |
|  |  |  | (−0.03888) | (−0.02766) |
| CELPE |  |  | −0.07264 · | −0.07264 ** |
|  |  |  | (−0.03888) | (−0.02766) |
| CEMIG |  |  | −0.04516 | −0.04516 |
|  |  |  | (−0.03888) | (−0.02766) |
| COPEL |  |  | −0.11873 ** | −0.11873 *** |
|  |  |  | (−0.03888) | (−0.02766) |
| ELETROPAULO |  |  | 0.01083 | 0.01083 |
|  |  |  | (−0.03888) | (−0.02766) |
| $R^2$ | 0.40062 | 0.69456 | 0.47853 | 0.77246 |
| Adj. $R^2$ | 0.39742 | 0.64554 | 0.45836 | 0.72579 |
| No. obs. | 189 | 189 | 189 | 189 |
| F statistic | 124.99126 | 14.16853 | 23.72758 | 16.54994 |
| $R^2$ | 0.40062 | 0.69456 | 0.47853 | 0.77246 |

Statistical significance indicated by *** $p < 0.001$, ** $p < 0.01$, * $p < 0.05$, · $p < 0.1$ and standard error in parentheses.

From the estimation in OLS in the before-and-after model, presented in four models (1, 2, 3, and 4) in Table 1, we analyzed the results, considering the fixed effect by date and fixed effect by company, which were simulated as follows: Model 1 (No and No); Model 2 (Yes and No); Model 3 (No and Yes), and Model 4 (Yes and Yes). The purpose of this study was to analyze the results of Model 4, as described in Section 4.2.

### 4.2. Analysis of the Estimation of Model 4 (OLS)

From the analysis of the results obtained with Model 4, we found that when introducing the company fixed effects, the estimation omitted some observations due to the existence of perfect multicollinearity. In this case, we found that the company CEEE and the years 1994 and 2018 were omitted, so their effects were captured by the intercept.

The estimated value for the contribution dummy was statistically significant and positive, indicating that the implementation of the contribution to innovation increased AST/MWh values. Thus, the null hypothesis was rejected, as a loss to consumers was identified after the implementation of the public policy.

The estimated values for the fixed effects showed that considering a 5% significance level as the base case for CEEE's 1994 tariffs, the tariffs of CELG, CELPE, and COPEL were lower than that of CEEE, while the tariffs of CEMIG, ELETROPAULO, and LIGHT were statistically equal to those of CEEE. In the same model, we found a positive effect on tariffs after the implementation of mandatory contributions to innovation (0.47592, exponential scale). Regarding the time-fixed effects, in certain periods, 2000 to 2003, 2008 to 2014, and in 2017, the tariff decreased, as statistically significant and negative values were found. Models 1, 2, and 3 produced results that corroborated the positive effect of innovation contribution on the tariff estimated in the model chosen for analysis. Model 4, however, produced results with different magnitudes, as shown in Table 2.

**Table 2.** Results of the estimated models.

|  | Model 1 | Model 2 | Model 3 | Model 4 |
|---|---|---|---|---|
| Interceptor | 5.84905 *** | 5.83802 *** | 5.88519 *** | 5.87416 *** |
|  | (−0.02325) | (−0.04367) | (−0.03367) | (−0.04247) |
| Dummy Contrib | 0.29471 *** | 0.47592 *** | 0.29471 *** | 0.47592 *** |
|  | (−0.02636) | (−0.06177) | (−0.02499) | (−0.05433) |
| Fixed effect by date | No | Yes | No | Yes |
| Fixed effect by company | No | No | Yes | Yes |
| $R^2$ | 0.40062 | 0.69456 | 0.47853 | 0.77246 |
| Adj. $R^2$ | 0.39742 | 0.64554 | 0.45836 | 0.72579 |
| Num. obs. | 189 | 189 | 189 | 189 |
| F statistic | 124.99126 | 14.16853 | 23.72758 | 16.54994 |

Statistical significance indicated by *** $p < 0.001$, ** $p < 0.01$, * $p < 0.05$, · $p < 0.1$ and standard error in parentheses.

When the goal was to examine only the effect of the contribution dummy and consider the company and time fixed effects as controls in the model, the results from Table 1 were obtained as shown in Table 2.

### 4.3. Behavior of AST/MWh (1994 to 2020) before and after ContribInova in Model 4

After the implementation of the SEB RD&I public policy in 2000, the evolution of the AST/MWh before and after the creation of ContribInova, which provides the financing of the program, needed to be verified. Therefore, we analyzed the behavior of the AST/MWh from 1994 to 2020.

The real AST/MWh was analyzed using the before-and-after model, where before was the period before the implementation of the SEB innovation public policy (1994 to 1999), and after was period after the implementation of the SEB innovation public policy (2000 to 2020). Therefore, the data analysis in the before-and-after model covered the period from 1994 to 2020. The data used were extracted from the literature [38].

Figure 4 shows the performance of the real energy tariff of the seven electricity distributors in the study sample.

The lowest AST/MWh in the period from 1994 to 2000 was for CEMIG, ranging from 306.10 to 351.19 BRL/MWh in present value (PV) for 2020, and the second-lowest was for LIGHT, with values ranging from 374.98 to 476.25 BRL/MWh, followed by CELG, CELPE, and COPEL. CEMIG's AST/MWh rapidly grew between 2001 and 2006 and became the highest between 2007 and 2016, but the position was lost to LIGHT as of 2016. LIGHT continues to lead the ranking of the highest actual AST/MWh in 2020 at 733.45 BRL/MWh. CEMIG's AST/MWh closed 2020 with a value of 684.12 BRL/MWh, having the second-highest value in the analysis group.

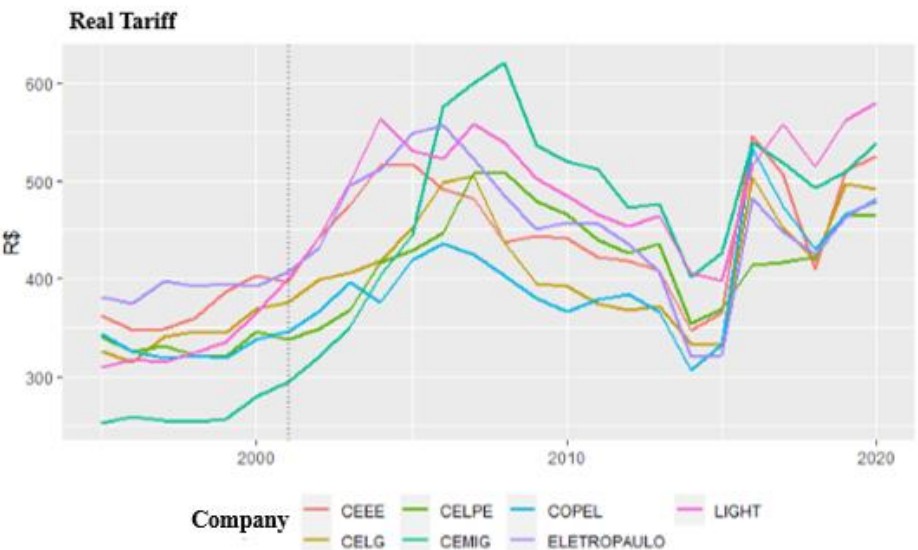

**Figure 4.** Electricity tariff (AST) (in BRL 1.00) from 1994 to 2020 in MWh.

Despite ANEEL's efforts to equalize the AST/MWh, a substantial difference persists between the lowest and highest tariffs in the Brazilian electricity market. This result is due to the inherent tariffs in regulated companies [42] because regulation struggles to address problems involving two important questions: Is the cost marginal cost or average cost? Who should cover the fixed costs: the user or the taxpayer? One school of thought holds that the user should pay the cost of the service they use, but some think differently and argue that the cost should be shared and paid, at least in part, by third parties who contribute to a service they do not use [42].

Figure 5 shows the behavior of the real AST/MWh before and after ContribInova, demonstrating the actual increase in AST/MWh that had already started to occur between 1994 and 1999 and continued in the following period, after the creation of the public policy. We adopted the base date of 1994, applying a zero-readjustment rate of AST/MWh, for all companies in the analysis group (zero-base). In 1994, the BRL was created; and in the month of the implementation of the URV and the new currency (BRL), the monthly inflation was approximately 85.0% per month.

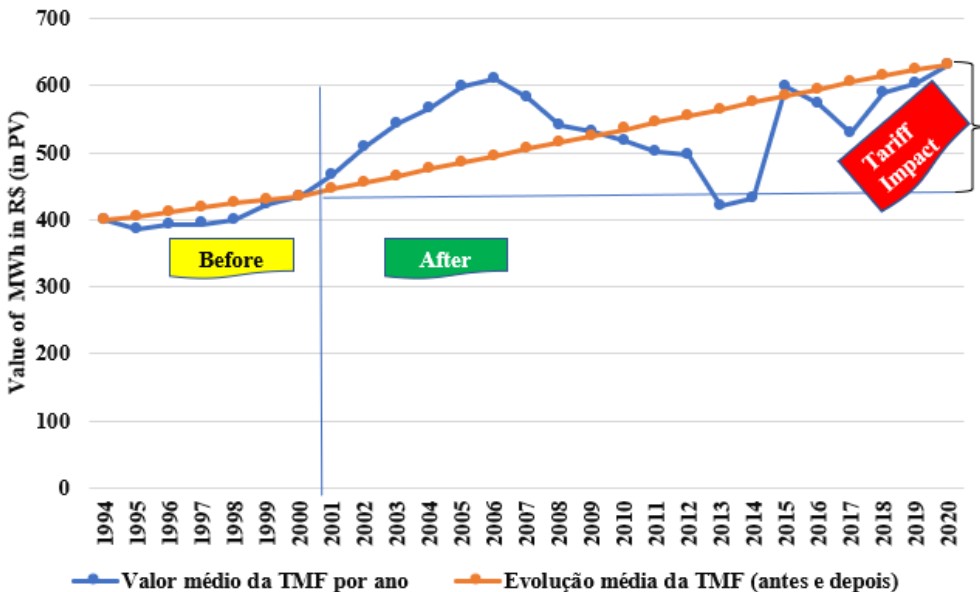

**Figure 5.** Evolution of the real AST from 1994 to 2020: MWh in BRL (in PV).

Price formation in the global electricity sector, including in Brazil, is complex. In Brazil, the sector involves a globalized supply chain; therefore, prices are subject to exchange rate variation. Added to these factors are the internal problems of the Brazilian economy (inflation and dependence on an international supply chain) associated with a complex generation, transmission, and distribution (GTD) network in a large country, where the large power generation projects are far from the large consumer centers [8,42].

SEB has the National Interconnected System (NIS) and can transport energy from the producing to the consuming regions and even import energy from other South American countries such as Argentina, Paraguay, and Uruguay and to meet the demand of different regions in the country. However, Brazil still experiences difficulties in finding a fair pricing methodology that meets the needs of society and the companies that operate in the SEB. This problem was termed by Tirole [42] as the regulation of access to the electricity transmission grid, which generates obstacles to the introduction of competition in Brazil and worldwide, including the U.S. and European Union electricity systems.

Thus, statistical regression analysis modeling, in combination with the executed AST/MWh projections, based on the IPCA for updating the PV values, enabled an adequate assessment of the impact of the SEB's RD&I public policy on the tariff moderation policy, which is in line with Agenda 2030 [1,19,43–47].

### 4.4. SEB's RD&I Public Policy versus Agenda 2030 SDG 7 Goal 7.1

The previously described results (Section 4.1–4.4) enabled discussions regarding the study objective, i.e., to evaluate the impact of the Brazilian RD&I public policy of the SEB on the average supply tariff (AST) in megawatt hours since its creation in 1998 and its implementation in 2000 until the present day. Notably, one of the aims of PD&I is tariff moderation, which is aligned with SDG 7, goal 7.1, of Agenda 2030 [10,28].

To compare the results of the SEB's RD&I public policy with the study objective, we evaluated the impact of the implementation of the SEB's RD&I public policy, with ContribInova being levied in the year 2000, using the "Before and After" model, which is one of the most widely used methods for public policy evaluation [1,46,47].

Additionally, we compared the AST/MWh to PV. We considered 1994 to 2020, separating the years into different periods: (a) 1994 to 2020, with the effect presented in Model 4, from 2000 to 2020, focusing on the AST/MWh value in BRL; (b) evaluation of the percentage variation in the AST/MWh for 1994 to 2020 and 2000 to 2020. To calculate the PV of the AST/MWh, the historical value was used, which is duly updated by the IPCA, which is the official inflation index in Brazil calculated by the IBGE [43,45,48].

The results produced by both models demonstrated an increase in the AST/MWh with the implementation of the public policy, suggesting possible contamination of the contribution created by law, which is levied on the net operating income (NOI) of electricity distribution companies, the object of this study. From the Model 4 (before and after regression) results, we detected a positive effect from the implementation of the innovation contribution (0.47592, on an exponential scale) in statistical/econometric language. Transformed into market language, this is equivalent to the AST/MWh increasing in real terms by about 60.95% in the period from 2000 to 2020 [44,49–51].

To understand the distortion caused by this increase, visualize a BRL 100.00 electric bill in 2000. Then, for the same bill today, for the same kilowatt hours, the cost would be BRL 160.95. This demonstrates the difference between the monetary update of a value and the real increase, which is explained by basic concepts of finance such as the value of money over time, including actual and present value [43,45].

A finance model was used as a complement to provide an alternative comparison with the results of the statistical/econometric model. However, we found the model also served to confirm that the model applied was correct because we noticed that the AST/MWh significantly increased as (a) CEMIG's AST/MWh increased (real increase) 94.80%; (b) the companies LIGHT, ELETROPAULO, and CELPE had AST/MWh increases of approximately 10.0% (9.95%, 10.18%, and 9.25%, respectively); (c) in the other companies,

a small increase occurred in AST/MWh: CEEE by 1.32%, COPEL by 2.45%, and CELG by 3.71% [43,45]. The AST/MWh that increased the most occurred at the end of the study (2000–2020) with a real increase of 6.671826% per year, equivalent to 288.19% over the 21 years.

The comparison of the two models corroborated the idea that the creation of the RD&I public policy burdened society, as directly reflected in the MWh price. This especially occurred in the period from 2000 to 2007, which was incorporated in the formation of the final price of electricity, regardless of moments of relief in 2020; prices returned to a high level again.

This has been happening when the International Monetary Fund (IMF) published the results of a study conducted in 195 countries, showing Brazil's economic robustness in terms of nominal GDP in dollars per purchase parity. However, important contradictions were also exposed, because Brazil showed a decrease in GDP per capita, especially when the economic power of the population was analyzed using the same methodology for different regions with different costs of living, a situation that worsened for lower income ranges [52]. In that study, Brazil ranked as the 8th largest economy in the world, which, with robust numbers, demonstrates its fragility by ranking 85th and should reach 90th in 2026, considering the GDP per capita in purchasing power parity according to the IMF study [52].

Recently, IPEA published in its Carta de Conjuntura, a warning about the impact of the increase in the electricity cost for the very-low- and low-income (income range denomination used by IPEA) strata of the population. The accumulated inflation over the last 12 months for this group of the population was 10.05%. For the high-income group, the accumulated inflation index was 7.11% for the same period [48,53].

We concluded that electricity prices are increasing at a time when the country is becoming poorer, with decreasing income and rising inflation. All this compromises the public policy of RD&I of the SEB, which has tariff moderation as one of its aims. Our findings show that the situation in reality is not in line with SDG 7 Goal 7.1 of Agenda 2030.

Given the poor results, we developed the electricity purchasing power index (EPP), which is presented in the following section.

### 4.5. Indicators for Electricity Purchasing Power Indicator (EPP)

Electricity is a basic commodity that affects the cost of living of families, the costs of companies, which are passed on to their products and services and impact the cost of services provided by the government (union, states, and municipalities) to citizens. Basically, the citizen ultimately pays the bill, hence the importance of thinking about and proposing the PCEE, which is an indicator of the purchasing power of electricity, focusing on the Brazilian income versus the amount paid in AST/MWh (without taxes).

Figure 6 shows the evolution of the three indicators, where the first two were calculated from 1994 to 2020, and the third was calculated from 2012 to 2020 because the indicator of the average income of Brazilians was released starting from 2012 [54]. Importantly, the indicators were prepared for a set of seven companies, the objects of this study. As 1994 was an atypical year due to the implementation of the Real Plan when the country experienced an inflation rate of 85% per month or approximately 1500% per year, we decided to analyze 1995 onward.

The results provided by the indicators were as follows: (a) EPPNMW started its historical series in 1995 with an index of 1.42 AST/MWh and reached 2020 at 1.72 MWh; (b) EPPMWBFB-DIEESE started in 1995 with 10.29 AST/MWh and arrived in 2020 with 8.74 MWh; (c) for EPPAIB-IBGE, from 2012, the year in which IBGE released the first version of the average income of Brazilians indicator, to 2020, the index increased from 4.38 MWh to 3.87 AST/MWh; (d) the $\overline{X_p}$ of EPP (of the three previous indicators), which opened its historical series in 1995, increased from 9.21 AST/MWh in 1994 to 6.58 AST/MWh in 2020.

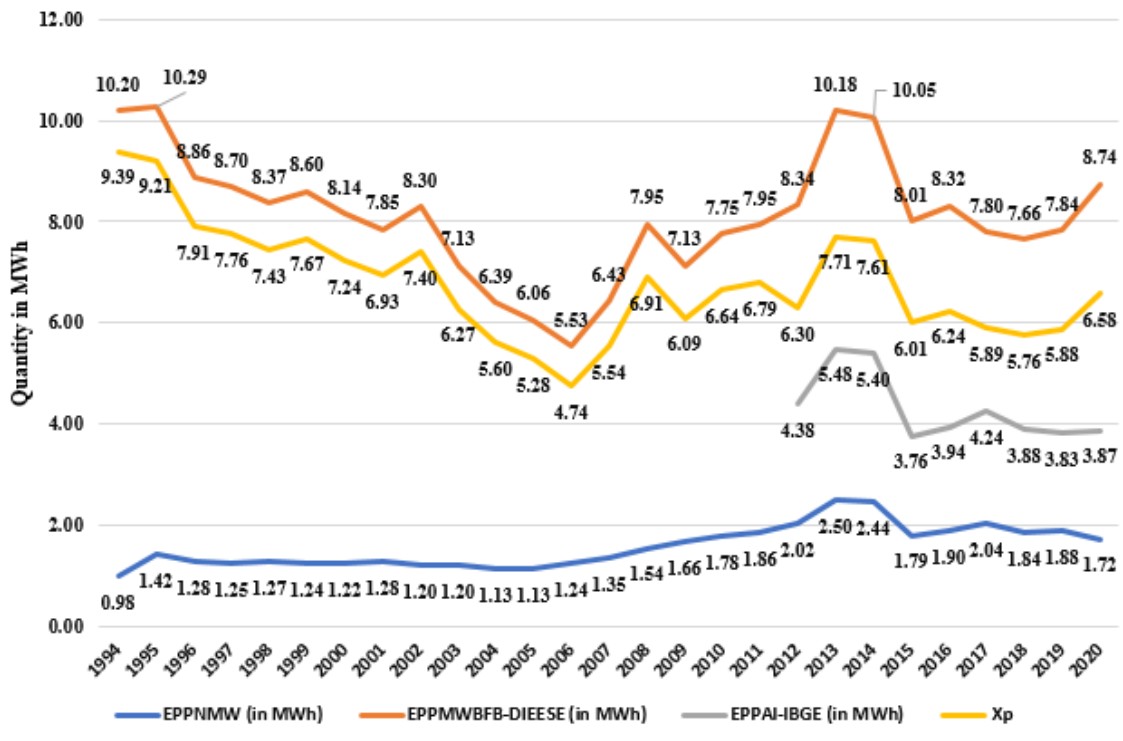

**Figure 6.** Evolution of electricity purchasing power indicators (1994–2020).

The results presented by the proposed indicators in the historical series from 1995 to 2020 showed that Brazilians lost purchasing power when comparing the relative income versus value of the AST/MWh, except for the PCEESMN indicator, which is based on the national minimum wage, which experienced a period with real gains (2003 to 2014). The arithmetic average of the three indicators also reflects the loss in purchasing power, again demonstrating Brazil's difficulty in fulfilling the Agenda 2030 agreement, which is to offer affordable electricity, as shown in Figure 6.

The $\overline{X}_p$ of EPP shows the average evolution of the three indicators in AST/MWh from 1995 to 2020. We generated a database to propose and calculate the electricity purchasing power index (EPP*i*), which is described below.

### 4.6. Proposed Electricity Purchasing Power Index (EPPi) from 2000 to 2020

Based on the proposed indicators, we decided to calculate the EPP*i* of the implementation of the SEB's RD&I public policy, in the year 2000. This allowed us to follow the tariff adjustment policy implemented in the sector by ANEEL. The EPP*i* was calculated from the $\overline{X}_p$ of the three EPP indicators (Xp of the EPP), in percentage base 100 (in 1999, the year before ContribInova was created). From 2000 on, the percentage variation was calculated, between the previous and the current year, up to the year 2020.

Figure 7 shows that from 2000 to 2020, AST/MWh increased more than the Brazilian population income. In the construction of the three EPP indicators and EPP*i*, the year 1999, the year before ContribInova started, was considered equal to 100.00% (the base year for the index). In 2000, the index was 94.31%; by 2006, it had reached 61.81%, showing an unequivocal loss of purchasing power of the electric energy consumer of approximately 38.19% in this period. As of 2007, a period of recovery of the purchasing power of electric energy began. In 2013, the EPP*i* was 100.45%, demonstrating that the consumer had an income slightly higher than the increase in electric energy in this year. However, in 2014 the EPP*i* started a new downward trend, reaching 75.08% in 2018, but rose again in 2020 to 85.82%, as shown in Figure 7.

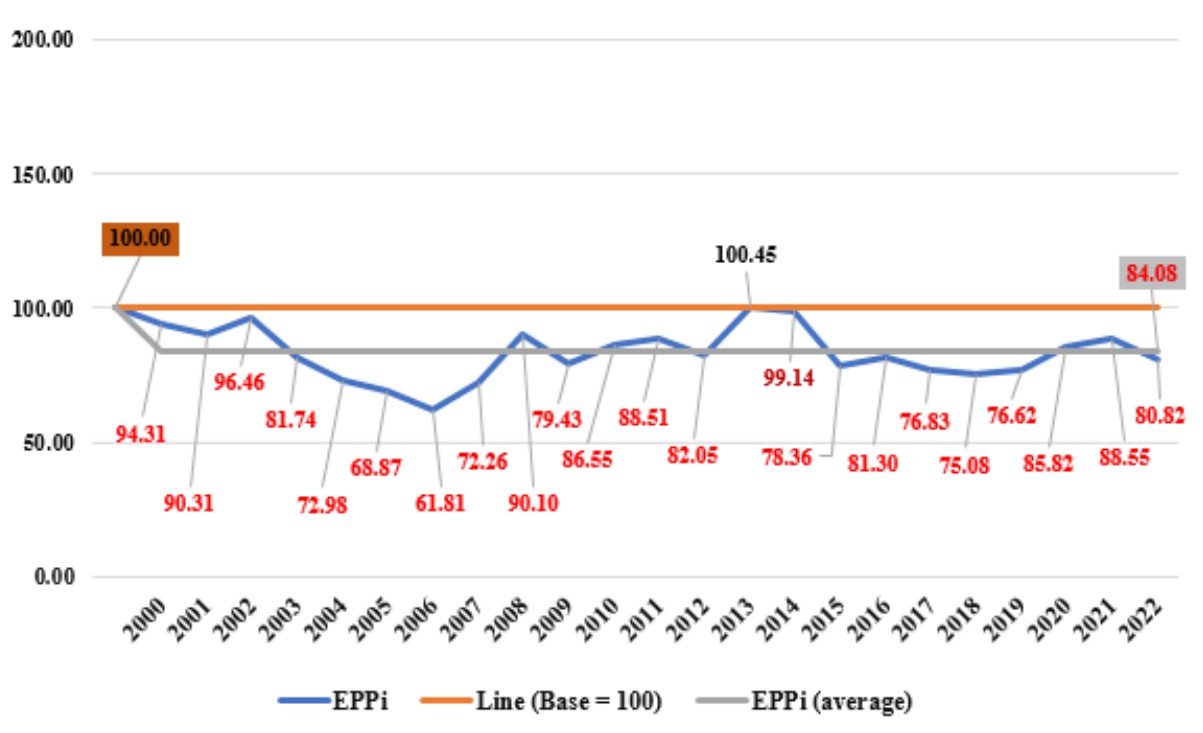

**Figure 7.** EPP*i* evolution from 2000 to 2020 and projected for 2021 and 2022.

As the EPP*i* oscillated during the 22 years (20 elapsed and 2 projected years), the average EPP*i* (weighted average) of the period was calculated to verify the average behavior of the index over time. The result was 84.08%, which confirmed a loss of purchasing power of electric energy from 2000 to 2022.

The tariff management process is dynamic and sensitive to periods of crisis, such as inflation, exchange rate policy changes, and water crises, especially the latter, because Brazil's electricity matrix is highly dependent on hydrological energy generation. In 2021, the Brazilian people faced a triple crisis (inflation, exchange rate, and drought), which has strongly impacted the price of electricity. Considering the increases applied in 2021, the AST/MWh increased approximately 8.00% and is expected to triple in 2022, reaching 27.00% [38].

As the present hydroelectric crisis is serious and may worsen according to data from ONS [55], we conducted projections considering the information released. Among this information are the increases in the AST/MWh by ANEEL [56] of the IPCA and the minimum wage by [48] correcting the minimum wage basic basket from DIEESE and maintaining the value of the average Brazilian income from 2020 to 2021 and 2022. With this, we obtained the 2021 and 2022 EPP*i* of 88.55% and 80.82%, respectively, showing the continuous fall in the electricity purchasing power. The situation is unfavorable for the consumer and may worsen if the water crisis is aggravated within the parameters projected by ONS [55].

The improvement in the EPP*i* from 2018 to 2020 is explained by the use of a highly aggregated indicator, that is, the IBGE's AIb, obtained through the IBGE wage mass study. The AIb markedly improved, because the crisis generated by COVID-19 increased unemployment in the salary ranges that receive the lowest wages, leaving fewer people employed but receiving higher wages. Thus, the salary mass decreased, but when divided by a smaller number of workers, a higher average income value was generated.

Finally, when analyzing the evolution of the EPP*i* over the period the RD&I public policy of the SEB was active (2000–2021), we found that this policy had practically no

impact on the reduction in the cost of electric energy services, despite this being one of its goals.

*4.7. Brazil versus International Electricity Prices*

When comparing the prices charged in Brazil with international standards, the situation is no different. A study by FIRJAN [57] showed that Brazil has the sixth most expensive tariff in the world at BRL 402.30 MWh, following the Czech Republic in fifth place with a value of BRL 408.90 MWh. Colombia is ranked fourth with BRL 414.10 MWh, Singapore in third at BRL 459.40 MWh, Italy in second place with BRL 536.10 MWh, and India ranked first at BRL 597.00 MWh. This same study showed that the electricity tariffs in Brazil are 46% higher than the international average [57].

In the comparison of the residential consumer electricity rates rankings, Brazil ranks 14 among 28 member countries of the International Energy Agency (IEA) according to a survey by the Brazilian Association of Electricity Distributors (Abradee) [58]. Among the BRICS countries, Brazil has the most expensive residential tariff [59].

The electricity price problem in Brazil crosses borders, indicating the need to pursue one of the goals of the RD&I public policy of the SEB, i.e., tariff moderation, as well as the commitments under the Paris Agreement, Agenda 2030, i.e., "by 2030 ensure universal access to affordable, reliable and modern energy services" [10,28].

*4.8. Consequences of High Electricity Prices in Brazil, Combined with Crises*

The Brazilian people are currently afflicted by a combination of an economic crisis aggravated by the coronavirus pandemic (COVID-19) and a water crisis, which has impacted the value of the energy bill in Brazil, as described Section 4.7. The successive price increases that have been occurring are above the average of recent years, mainly as a result of the introduction of the water scarcity flag. The constant and growing increases have caused an increase in nonpayment, which, before the COVID-19 pandemic, was around 3.35% but reached 6.38% between September and December 2021 in the residential segment [30].

ANEEL has taken a socially relevant measure by determining, on 1 April 2022, that 11.3 million families nationwide should be automatically included in the Electricity Social Tariff program (Law 12.212/2010, the legal framework of the Social Tariff for Electrical Energy, which grants a discount on the electricity bill for low-income families, including families that are part of the Unified Registry and the Continuous Cash Benefit (BPC)), which, when added to the current 12.4 million, should total approximately 23 million [31]. This decision will impact nonpayment and prevent the cutting off of the electricity supply to these families [29–31].

## 5. Final Considerations

The aim of this study was to evaluate the impact of the implementation of the SEB's RD&I public policy on the tariff moderation policy provided for in the innovation program regulated by ANEEL, which has been active for 21 years. Additionally, with the advent of the Paris Agreement and the creation of the UN Agenda 2030, Brazil has committed to the SDGs, in particular SDG 7, goal 7.1, which aims to ensure universal, reliable, modern, and affordable access to energy services. As such, our focus in this study was to specifically analyze the affordability of energy services.

Our study has relevant contributions, providing an opportunity for reflection in the academic community, companies that operate in the SEB, public managers, and Brazilian society in general, since:

First, we found that both the premise of tariff moderation of SEB and the commitment to Agenda 2030 to offer energy and services at affordable prices are not yet a reality for the Brazilian people.

Second, the previous statements are based on two results from this study: (a) The introduction of ContribInova to finance the RD&I of the SEB indicates strong evidence of a real increase in the electric energy tariff soon after the implementation of the public

policy, because it generated a real increase of approximately 60.95% from 2000 to 2020, according to Section 4.4. (b) The evolution of the proposed energy purchasing power index (EPP*i*, base 100) demonstrated that in the 22 years of the RD&I of the SEB, the Brazilian population income (minimum wage, minimum wage from DIEESE, and average Brazilian income from IBGE) lost to the AST practically every year, except for 2013. The average in the period was 84.8%. Every time the EPP*i* is below 100, income is at a disadvantage concerning the price of electricity.

Third, the results of the comparison of the electricity tariff prices in Brazil with those in other countries confirmed that Brazil's position is serious because (a) Brazil is ranked sixth regarding the most expensive electricity; (b) regarding to the residential tariff, Brazil ranks 14 among 28 member countries of the International Energy Agency; and (c) among the BRICS, Brazil has the most expensive residential electricity tariff.

Finally, we identified an opportunity for research that could directly impact the price of electricity, the tax issue, which was not addressed in this study because this topic is complex and should be addressed in a separate study. We need to understand the structure of the Brazilian electricity bill, which is generally as follows: taxes represent 42.1% of the bill, the cost of electricity accounts for 39.7%, the companies' remuneration for 15.5%, and 2.7% are transmission costs. Understanding and proposing alternatives that reduce tax costs, the cost of energy, and the services involved are fundamental to reducing the total electricity bill cost in Brazil, thus enabling the benefits of innovation to reach the end consumer.

**Author Contributions:** Conceptualization, G.d.S.M.; Methodology, G.d.S.M. and M.A.d.P.D.; Validation, M.A.d.P.D. and J.L.d.A.F.; Formal analysis, G.d.S.M. and M.A.d.P.D.; Investigation, G.d.S.M.; Writing – original draft, G.d.S.M.; Writing – review & editing, G.d.S.M., M.A.d.P.D. and J.L.d.A.F.; Supervision, M.A.d.P.D. and J.L.d.A.F. All authors have read and agreed to the published version of the manuscript.

**Funding:** This research received no external funding.

**Informed Consent Statement:** Not applicable.

**Data Availability Statement:** Not applicable.

**Acknowledgments:** The publication of this study was made possible through financial aid granted by Fundação de Apoio à Pesquisa do Distrito Federal (FAPDF), according to process No. 00193-00000961/2023-62.

**Conflicts of Interest:** The authors declare no conflict of interest.

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
