# Peer review of "Public Innovation Policy Evaluation: Brazilian Electricity Tariff Paid by Consumers"

_sustainability, doi:10.3390/su151310142_

Round 1
Reviewer 1 Report
I am interested in what is new in this paper, and what has not been previously published in the author's doctoral thesis, at the 3rd SDEWES LA Conference in São Paulo, Brazil (July 24-28, 2022) and in the paper of the identical name, which has already been published in the "SSRN Electronic Journal" in January 2022?? The proposed article is completely self-plagiarized! The article must therefore be rejected!
Author Response
Dear Reviewer 1,
I inform you that the questions raised by you. were replied to Ms Sierra, so nothing to add.
Thanks!
Gilmar Dos Santos Marques
corresponding author

Reviewer 2 Report
The author has provided significant novelty and following correction need to incorporate before publication
Title does not suit the content; it needs to rephrase.
IMPACT EVALUATION OF THE PUBLIC INNOVATION 1 POLICY OF THE BRAZILIAN ELECTRICITY SECTOR (SEB) 2 ON THE ELECTRICITY TARIFF PAID BY THE CONSUMER
Not short form in the title i.e. (SEB)
There are many Grammatic, and typo error are like "Be-fore and After Real Tariff"
Figure 2 to Figure 8 are not cited in text
Author Response
Dear Reviewer 2,
Thank you for your contributions to our work.
As for the title, we chose to change it, following your suggestion, and from now on it will be as follows: “Public Innovation Policy Evaluation: Brazilian Electricity Tariff Paid by Consumers”
The acronym “SEB” only refers to “SETOR ELÉTRICO BRASILEIRO”, but we removed it from the title.
The revision of the English language was carried out by the MDPI English Editing service and we believe that it will fully meet the demand, as per the attached file.
With regard to the organization of figures, as well as citations in the text were duly adjusted.
Best regards,
Gilmar Dos Santos Marques
corresponding author

Reviewer 3 Report
This paper evaluates the impact of SEB policy of innovation, average supply tariff (AST) during 2000 to 2020 considering premise of tariff moderation as foreseen in RD&I of the SEB. Study is well-designed and well presented in the paper. This paper addresses an import research question of "impact of the implementation of the SEB's RD&I public policy on the tariff moderation policy provided for in the innovation program regulated by ANEEL, in execution for the last 21 years." The presentation is clear and concise, and the data are well-presented and analyzed. The abstract and introduction is presented in a informative manner and provides a clear background to the study. The result analysis section is well-written, and the appropriate mathematical methods are applied. The discussion section is well-written and provides a clear interpretation of the findings. The references are not mostly up-to-date and however they are relevant to the study. However, there are a few places where more recent studies could be cited to support the findings. Overall, the manuscript can be accepted with some additional literature review (2020, 2021,2022). If the above suggestions are addressed, it improves quality of the manuscript.
Author Response
Dear Reviewer 3
Thank you for your contributions to our article.
To meet your suggestion regarding the literature review, a re-reading was carried out with the inclusion of new references, as per the attached file.
The bibliographic references used in the article total 59 (fifty-nine), which were distributed throughout the text, in the form of citations, as follows:
a) Citations referring to the period from 2017 to 2023: totaling 35 citations, distributed in the Introduction: 5; Theoretical Reference: 17; Methodological Procedures: 13. This period corresponds to 59.3%
b) Citations prior to 2017: totaling 24 citations. This period corresponds to 40.7%.
As the study is long-lived, as it analyzes data from a period of 20 years, we believe that the proportion of citations of 60% of bibliography from the last 5 years and 40% of sources prior to this period, causes an adequate balance of references. .
Best regards,
Gilmar Dos Santos Marques
corresponding author
